# Online Linear Classification with Massart Noise

**Ilias Diakonikolas** [1]  **Vasilis Kontonis** [2]  **Christos Tzamos** [3]  **Nikos Zarifis** [1]

## Abstract

We study the task of online learning in the presence of Massart noise. Specifically, instead of assuming that the online adversary chooses an arbitrary sequence of labels, we assume that the context $x$ is selected adversarially but the label $y$ presented to the learner disagrees with the ground-truth label of $x$ with unknown probability *at most* $\eta$. We focus on the fundamental class of $\gamma$-margin linear classifiers and present the first computationally efficient algorithm that achieves mistake bound $\eta T + o(T)$. We point out that the mistake bound achieved by our algorithm is qualitatively tight for computationally efficient algorithms; this follows from the fact that, even in the offline setting, achieving 0-1 error better than $\eta$ requires super-polynomial time under standard complexity assumptions.

We extend our online learning model to a $k$-arm contextual bandit setting where the rewards—instead of satisfying commonly used realizability assumptions—are consistent, in expectation, with some linear ranking function with weight vector $w^*$. Given a list of contexts $x_1, \ldots x_k$, if $w^* \cdot x_i > w^* \cdot x_j$, the expected reward of action $i$ must be larger than that of $j$ by at least $\Delta$. We use our Massart online learner to design an efficient bandit algorithm that obtains expected reward at least $(1 - 1/k) \Delta T - o(T)$ bigger than choosing a random action at every round.

## 1. Introduction

Online prediction has a rich history dating back to the works of (Robbins, 1951; Hannan, 1957; Blackwell et al., 1954).

---
*Equal contribution [1]Department of Computer Sciences, University of Wisconsin-Madison, Madison, USA [2]Department of Computer Science, University of Texas-Austin, Austin, USA [3]Department of Computer Science, University of Athens, Athens, Greece. Correspondence to: Vasilis Kontonis <vasilis@cs.utexas.edu>, Nikos Zarifis <zarifis@wisc.edu>.

*Proceedings of the $42^{nd}$ International Conference on Machine Learning*, Vancouver, Canada. PMLR 267, 2025. Copyright 2025 by the author(s).

In the online scenario, the learner's objective is to tackle a prediction task by acquiring a hypothesis from a series of examples presented one at a time. The aim is to minimize the overall count of incorrect predictions, known as the mistake bound, considering the knowledge of correct answers to previously encountered examples (Littlestone, 1988; 1989; Blum, 1990; Littlestone & Warmuth, 1994; Maass & Turan, 1994). In the context of online linear classification, i.e., when the presented labels can be realized by a linear threshold function, the seminal perceptron algorithm (Rosenblatt, 1958; Novikoff, 1962), was the first online learning algorithm.

**Realizable and Agnostic Online Learning**  In (Littlestone, 1988) the realizable online classification setting was defined, where the adversary is allowed to select an arbitrary datapoint $x^{(t)}$ at every round but the label $y^{(t)}$ must be consistent with an underlying ground hypothesis $f$ from a class $\mathcal{H}$. The number of mistakes in the realizable setting was shown to be characterized by the Littlestone dimension $\mathrm{LD}(\mathcal{H})$ of the class $\mathcal{H}$. Similarly to the agnostic PAC learning of (Haussler, 1992), in (Ben-David et al., 2009), motivated by the fact that often the observed labels are noisy, the setting of online learning with label noise was introduced. In the most extreme case of agnostic (adversarial) label noise where no assumptions are placed on the labels, it was shown that the regret over $T$ rounds is $\widetilde{O}(\sqrt{T \, \mathrm{LD}(\mathcal{H})})$. Even though the regret in the agnostic setting was shown to be sublinear in $T$, the corresponding computational task is far from being well-understood. In particular, even for simple classes such as linear classifiers efficient online algorithms with sublinear regret would imply efficient algorithms in the offline setting – a well known computationally intractable problem (Guruswami & Raghavendra, 2006).

**Online Learning with Massart Noise**  Investigating regimes beyond the above, worst-case, agnostic setting has been an important area of research with the goal to get improved regret (and mistake) bounds but also to allow the design of efficient algorithms. Similar to the definition of Massart or Bounded noise (Massart & Nedelec, 2006) for offline (PAC) learning, in (Ben-David et al., 2009) a *semi-random* online classification setting was introduced with a focus on improving over the agnostic regret bounds. In this model, while the online adversary is allowed to pick

arbitrary locations to present to the learner, the labels that they choose must be consistent with the ground-truth with probability strictly larger than 50%.

**Definition 1.1** (Online Learning with Massart Noise (Ben–David et al., 2009))**.** Fix a class of concepts $\mathcal{C}$ over $\mathbb{R}^d$ and a target concept $c^* \in \mathcal{C}$. An (oblivious) adversary selects a sequence of examples $\boldsymbol{x}^{(1)}, \ldots, \boldsymbol{x}^{(T)}$ and noisy label random variables $y^{(1)}, \ldots, y^{(T)}$ such that for all $t$ it holds that $\mathbf{Pr}[y^{(t)} \neq c^*(\boldsymbol{x}^{(t)}) \mid \boldsymbol{x}^{(t)}] \leq \eta$. At round $t = 1, \ldots, T$, the learner observes $\boldsymbol{x}^{(t)}$, predicts a label $\hat{y}^{(t)}$, and suffers loss $\mathbb{1}\{\hat{y}^{(t)} \neq y^{(t)}\}$. The goal of the learner is to minimize the total number of mistakes, defined as $M(T) = \mathbf{E}\left[\sum_{t=1}^{T} \mathbb{1}\{y^{(t)} \neq \hat{y}^{(t)}\}\right]$.

*Remark* 1.2 (Regret vs Mistake Bound). In Definition 1.1, often the regret over $T$ rounds is used, i.e., the expected difference between the mistakes made by the learner and the minimum number of mistakes that any hypothesis $h$ in $\mathcal{H}$ could achieve

$$R(T, h) = M(T) - \mathbf{E}\left[\sum_{t=1}^{T} \mathbb{1}\{y^{(t)} \neq h(\boldsymbol{x}^{(t)})\}\right].$$

*Remark* 1.3 (Random vs Massart Noise). We stress that the probability that a label is flipped in Definition 1.1 is *at most* $\eta$ (and not equal to $\eta$). The fact that the flip probability is not uniformly $\eta$ for all examples results in asymmetric noise, which is exactly what makes (even offline) learning with Massart noise algorithmically challenging. In fact, as we point out in Remark 1.10, achieving optimal error (or regret) in the Massart setting is computationally hard (which is not the case when the labels are flipped uniformly with probability $\eta$). Here we will focus on obtaining computationally efficient online algorithms matching the best-known offline guarantees of (Diakonikolas et al., 2019).

In (Ben-David et al., 2009), an algorithm based on Littlestone's Standard Optimal Algorithm is given that achieves a mistake bound $M(T) \leq \min_{h \in \mathcal{H}} \sum_{t=1}^{T} \mathbb{1}\{h(\boldsymbol{x}^{(t)}) \neq y^{(t)} + \mathrm{LD}(\mathcal{H}) \log T / (1 - 2\sqrt{\eta(1-\eta)})$. Therefore, as long as $\eta$ is bounded away from $1/2$, a strong separation between the agnostic regret (that grows roughly as $\Omega(\sqrt{T})$) and the Massart regret was established. However, as discussed in (Ben-David et al., 2009), this and other similar approaches for establishing regret bounds do not yield computationally efficient algorithms as they rely on computing combinatorial quantities that are at least as hard as computing the VC-dimension (Frances & Litman, 1998).

**Linear Classification** To investigate the computational aspects of online classification we focus on the fundamental class of linear classifiers or halfspaces, i.e., functions of the form $h(\boldsymbol{x}) = \mathrm{sign}(\boldsymbol{w} \cdot \boldsymbol{x})$ for some weight vector $\boldsymbol{w} \in \mathbb{R}^d$. In the offline setting, a long line of recent works (Awasthi et al., 2015; Diakonikolas et al., 2019; 2020; Chen

et al., 2020; Diakonikolas et al., 2021a;b) has successfully bypassed the computational hardness of worst-case agnostic learning and has provided efficient algorithms for linear classification in the presence of Massart noise. However, no efficient algorithm that achieves any non-trivial mistake bound is known for the more challenging online Massart setting of Definition 1.1. In this work, we aim to answer the following fundamental question.

*Are there* computationally efficient *algorithms for online linear classification with Massart noise?*

**Massart Bandits** Going beyond full-information online learning, we consider the multi-armed bandit setting: a multi-disciplinary research area that was initiated by (Thompson, 1933) and has received enormous attention in the past 30 years (see (Berry & Fristedt, 1985; Cesa-Bianchi & Lugosi, 2006; Slivkins et al., 2019) and references therein). In this work, we focus on the $k$-arm contextual bandit setting, where at every round $t$ the learner receives $k$ contexts $\boldsymbol{x}_1^{(t)}, \ldots \boldsymbol{x}_k^{(t)}$, chooses an action $\alpha = 1, \ldots, k$, and receives the reward/loss of their chosen action (the rewards of the other actions are not revealed). In this setting a common assumption known as realizability (Filippi et al., 2010; Abbasi-Yadkori et al., 2011; Chu et al., 2011; Agarwal et al., 2012; Li et al., 2017; Foster et al., 2018; Foster & Rakhlin, 2020) prescribes that the expected rewards are parametric functions of the contexts, e.g., $\mathbf{E}[r_i \mid \boldsymbol{x}^{(t)}] = \boldsymbol{w} \cdot \boldsymbol{x}_i^{(t)}$ for some unknown weight vector $\boldsymbol{w}$. Using this structural assumption, those works typically reduce the bandit problem to an online regression problem which can often be solved efficiently (for linear and generalized linear models). An orthogonal direction (Langford & Zhang, 2007; Dudik et al., 2011; Agarwal et al., 2014) makes minimal distributional assumptions and reduces the contextual bandit problem to agnostic (offline) classification. Since agnostic classification is computationally intractable (for most nontrivial classes) such approaches do not provide end-to-end efficient algorithms and rely on heuristics to solve the underlying classification task. Therefore, existing algorithms for contextual bandits either (i) make strong realizability assumptions and reduce the problem to (linear) regression or (ii) make minimal assumptions but face the computational intractability of agnostic classification. Motivated by the online Massart classification setting of (Ben-David et al., 2009), in this work, we propose a semi-random and semi-parametric Bandit model that lies between those two extreme settings and investigate the design of "end-to-end" efficient algorithms.

Before we describe our semi-random noise model, we start with its "noiseless" version. Instead of assuming that the expected rewards are given by a linear function of the contexts, we assume that they are *ranked* according to their linear

scores, i.e., if $\boldsymbol{w}^* \cdot \boldsymbol{x}_i > \boldsymbol{w}^* \cdot \boldsymbol{x}_j$ then the reward of action $i$ must be at least as that of action $j$.

**Definition 1.4** (Contexts and Linearly Sorted Rewards). Let $\mathcal{B}$ be the $d$-dimensional unit ball. We define $\mathcal{X} = \{(\boldsymbol{x}_1, \ldots, \boldsymbol{x}_k) : \boldsymbol{x}_1, \ldots, \boldsymbol{x}_k \in \mathcal{B}\}$ to be the context space. For simplicity, we view each context as a $d \times k$ matrix $\mathbf{X}$. Fix $M > 0$ and let $\boldsymbol{w}^* \in \mathbb{R}^d$ be some unit vector. Given a context $(\boldsymbol{x}_1, \ldots, \boldsymbol{x}_k)$, we say that a reward vector $\boldsymbol{r} \in [0, M]^k$ is sorted according to $\boldsymbol{w}^*$ if

$$\text{for all } i, j : \quad (\boldsymbol{r}_i - \boldsymbol{r}_j)(\boldsymbol{w}^* \cdot \boldsymbol{x}_i - \boldsymbol{w}^* \cdot \boldsymbol{x}_j) \geq 0\,.$$

Under the linear ranking setting of Definition 1.4 one can reduce the bandit problem to a noiseless linear classification task and design efficient algorithms (based on perceptron or linear programming) that will eventually learn to select the action with the highest reward. Our semi-random model extends the above definition where we require that rewards are only sorted in expectation with some margin $\Delta > 0$.

**Definition 1.5** (Monotone Reward Distributions). Let $\mathcal{X}$ be the context space and $\mathcal{T} = \mathbb{N}$ be the set of rounds. Fix some unit vector $\boldsymbol{w}^* \in \mathbb{R}^d$, $M > 0$, and $\Delta > 0$. Define a class of distributions $\mathcal{D}$ indexed by rounds $t$ and contexts $\mathbf{X}$: $\mathcal{D} = \{D^{(t, \mathbf{X})} : t \in \mathcal{T}, \mathbf{X} \in \mathcal{X}\}$. We assume that each reward distribution $D^{(t, \mathbf{X})}$ is supported on $[0, M]^k$. We say that the class $\mathcal{D}$ has monotone rewards with respect to $\boldsymbol{w}^*$ with margin $\Delta$ if for all $t \in \mathcal{T}$, $\mathbf{X} \in \mathcal{X}$, $i, j \in [k]$ with $i \neq j$ it holds

$$\mathop{\mathbf{E}}_{\boldsymbol{r} \sim D^{(t, \mathbf{X})}}[\boldsymbol{r}_i - \boldsymbol{r}_j \mid \boldsymbol{w}^* \cdot \boldsymbol{x}_i > \boldsymbol{w}^* \cdot \boldsymbol{x}_j] \geq \Delta\,.$$

*Remark* 1.6 (Massart Online Classification as a 2-arm Bandit). To obtain the online Massart setting we set the rewards $\boldsymbol{r}^{(t)} = ((1 + y^{(t)})/2, (1 - y^{(t)})/2)$ and the contexts $\mathbf{X}^{(t)} = (\boldsymbol{x}^{(t)}, -\boldsymbol{x}^{(t)})$. We observe that in that case when $y^{(t)}$ satisfies the Massart noise condition of Definition 1.1, it holds that conditional on $\boldsymbol{w}^* \cdot \boldsymbol{x} > 0$ it holds that $\mathbf{E}[(\boldsymbol{r}_1^{(t)} - \boldsymbol{r}_2^{(t)})] \geq 1 - 2\eta$. Therefore, Definition 1.5 is satisfied with $\Delta = 1 - 2\eta$.

We stress that the monotone reward distributions of Definition 1.5 are a semi-parametric model as we do not assume that the expected rewards have some parametric form rather than only require that they are sorted with respect to a linear sorting function. A similar semi-random linear sorting model was used in (Fotakis et al., 2022) in the context of (offline) learning linear rankings with bounded noise under the Gaussian distribution. We next define our contextual bandit model.

**Definition 1.7** (Contextual Bandits with Monotone Rewards (Massart Bandits)). Fix some unit vector $\boldsymbol{w}^* \in \mathbb{R}^d$, $\gamma, \Delta, M > 0$, and a class of monotone reward distributions $\mathcal{D}$ (see Definition 1.5). At round $t$:

1. The adversary picks context $\mathbf{X}^{(t)} \in \mathcal{X}$ and draws a random reward vector $\boldsymbol{r}^{(t)}$ from $D^{(t, \mathbf{X}^{(t)})}$.

2. The learner observes the context $\mathbf{X}^{(t)}$, picks action $a^{(t)} \in [K]$ and receives reward $\boldsymbol{r}_{a^{(t)}}$.

We also define the full-information setting the same way except from the last step where the learner receives reward $\boldsymbol{r}_{a^{(t)}}$ and observes the full reward vector $\boldsymbol{r}^{(t)}$.

*Remark* 1.8. *In what follows, we shall often simplify notation by writing $D^{(t)}$ instead of $D^{(t, \mathbf{X})}$ for the reward distribution at round $t$.*

### 1.1. Our Results

Our first result answers our main question posed in Section 1 and gives an efficient online classification algorithm that makes roughly $\eta T + o(T)$ mistakes in the Massart noise model of Definition 1.1. Our algorithm only requires that the sequence of examples picked by the adversary satisfies a standard $\gamma$-margin assumption. Without the margin assumption it is known (Littlestone, 1988) that, even in the noiseless setting, it is information theoretically impossible for the learner to do less than $T$ mistakes.

**Theorem 1.9.** *Consider the Online Massart Learning setting of Definition 1.1. Additionally, assume that the examples picked by the adversary have at least $\gamma$-margin with respect to some target halfspace, i.e., for all $t = 1, \ldots, T$, it holds that $\|\boldsymbol{x}^{(t)}\|_2 \leq 1$ and $|\boldsymbol{w}^* \cdot \boldsymbol{x}^{(t)}| \geq \gamma$, for some unit vector $\boldsymbol{w}^*$. There exists an algorithm that does $M(T) = \eta T + O(T^{3/4}/\gamma)$ mistakes and runs in $\mathrm{poly}(d)$ time per round.*

Our mistake bound matches (when viewed as an offline PAC learning result) the best known error guarantees of the corresponding offline learners of Massart halfspaces with margin given in (Diakonikolas et al., 2019; Chen et al., 2020). Moreover, the mistake-bound achieved by our algorithm is essentially best-possible when considering computationally efficient (statistical query) algorithms: in the recent works (Diakonikolas & Kane, 2020; Diakonikolas et al., 2022; Nasser & Tiegel, 2022) that consider offline (PAC) learning with Massart noise, it is shown that, even when the underlying distribution has $\gamma$-margin, no polynomial-time algorithm can achieve classification error better than $\eta/\mathrm{polylog}(1/(1 - 2\eta))$ in the Statistical Query framework. By a standard online to offline reduction this readily implies a $\eta T$ lower bound (up to $\mathrm{polylog}(1/(1 - 2\eta))$ factors).

*Remark* 1.10 (Information-Theoretic vs Computationally Efficient Online Learning). The $\eta T$ mistake bound *is not information-theoretically optimal*: in particular, in (Ben-David et al., 2009)) better, near-optimal, mistake bounds are given albeit with inefficient algorithms (i.e., with run-time exponential in the dimension $d$). When considering

computationally efficient algorithms, as we do here, there is strong evidence (see the SQ lower bounds of (Diakonikolas & Kane, 2020; Diakonikolas et al., 2022)]) that the $\eta T$ mistake is essentially best possible. Before our work no computationally efficient algorithm was known that could beat the random guessing benchmark (that makes $T/2$ mistakes). Moreover, we remark that, even in the offline setting of linear classification under Massart noise, the first computationally efficient that achieved classification error $\eta$ (the offline equivalent of $\eta T$ mistakes) was only given in the relatively (given the history of the problem) recent work (Diakonikolas et al., 2019).

Our algorithm is particularly simple: we perform online gradient descent on a sequence of reweighted Leaky-ReLU loss functions. The Leaky-ReLU loss has been successfully used in several works on learning with Random Classification and Massart noise (Bylander, 1998; Diakonikolas et al., 2019; Chen et al., 2020). In the online setting however simply using (online) gradient descent on the Leaky-ReLU does not suffice: even though the adversary is restricted to select examples with margin with respect to some ground-truth halfspace, they can still select examples very close to the decision boundary of the current hypothesis which would cause online gradient descent to get stuck or converge to sub-optimal solutions. To overcome this issue, we reweight the Leaky-ReLU loss by the margin of the current example according to the current hypothesis vector, see Algorithm 1. We are then able to show that standard regret guarantees for Online Convex optimization can be translated to obtain mistake bounds in the presence of Massart noise, see Lemma 2.2. Finally, we remark that our technique is also valuable for the offline setting: the recent works (Chandrasekaran et al., 2024; Diakonikolas & Zarifis, 2024), building on the ideas of this paper, were able to design sample-optimal algorithms for *offline* Massart classification.

We next present our result on semi-random "Massart" $k$-arm setting presented in Definition 1.5. In addition to the $\Delta$ "reward-margin" assumption of Definition 1.5, similarly to our online classification result of Theorem 1.9, we require a $\gamma$ "geometric-margin" assumption for the contexts with respect to some halfspace. We give an efficient bandit algorithm, see Algorithm 3, that is able to accumulate expected reward roughly $\Delta T$ more than playing a random action at every round.

**Theorem 1.11** (Monotone $k$-arm Contextual Bandits). *Consider the monotone reward online setting of Definition 1.5. Moreover, for some unit vector $\boldsymbol{w}^* \in \mathbb{R}^d$, assume that for all $t$, it holds that for all $i$ $\|\boldsymbol{X}_i^{(t)}\|_2 \leq 1$ and for all $i \neq j$, $|\boldsymbol{w}^* \cdot \boldsymbol{X}_i^{(t)} - \boldsymbol{w}^* \cdot \boldsymbol{X}_j^{(t)})| \geq \gamma$. There exists a bandit algorithm that runs in $\mathrm{poly}(d)$ time per round and selects a sequence*

*of arms $\alpha^{(1)}, \ldots, \alpha^{(T)} \in [k]$ that obtain expected reward*

$$\mathbf{E}\left[\sum_{t=1}^T \boldsymbol{r}^{(t)}(\alpha^{(t)})\right] \geq \mathbf{E}\left[\sum_{t=1}^T \frac{1}{k} \sum_{i=1}^k \boldsymbol{r}_i^{(t)}\right] + \frac{k-1}{k}\Delta T - O(T^{5/6}(k\Delta M^2)^{1/3}/\gamma).$$

Our algorithm for the $k$-arm bandit setting relies on the observation that, assuming that we could observe all rewards $r_i^{(t)}$, then one could treat the labeled pairs $(\boldsymbol{x}_i^{(t)} - \boldsymbol{x}_j^{(t)}, r_i^{(t)} - r_j^{(t)})$ as real-valued versions of online linear classification with Massart noise and provide it as input to our online learning algorithm. As is common in bandit problems, to adapt this "full-information" approach to the bandit setting, we pick a random action with small probability at every round that provides us with unbiased estimates of the full reward vectors. For more details we refer to Section 3. Finally, we remark that for the special case of 2-armed bandits our result implies the following corollary.

**Corollary 1.12** (Monotone 2-arm Contextual Bandits). *Consider the monotone reward online setting of Definition 1.5. In the bandit setting, Algorithm 3 produces a sequence of arm choices $\alpha^{(1)}, \ldots, \alpha^{(T)} \in \{1, 2\}$ that obtain expected reward*

$$\mathbf{E}\left[\sum_{t=1}^T \boldsymbol{r}^{(t)}(\alpha^{(t)})\right] \geq \mathbf{E}\left[\sum_{t=1}^T \frac{\boldsymbol{r}_1^{(t)} + \boldsymbol{r}_2^{(t)}}{2}\right] + \frac{1}{2}\Delta T - O((M^2\Delta)^{1/3}T^{5/6}/\gamma).$$

Corollary 1.12 is a generalization of our online learning result of Theorem 1.9: indeed, using Remark 1.6, we obtain that the expected reward is equal to $T - M(T)$, where $M(T)$ are the expected mistakes. Thus, Corollary 1.12 implies that $T - M(T) \geq T/2 + (\Delta/2)T + o(T)$ and, using the fact that $\Delta = 1 - 2\eta$ and $M = 1$, we get that $M(T) \leq \eta T + o(T)$. Given the hardness result for improving upon $\eta T$ regret for online Massart classification that we already discussed we conclude that our reduction from Bandit to Online Massart is tight for 2-armed bandits.

### 1.2. Notation

For $n \in \mathbb{Z}_+$, let $[n] := \{1, \ldots, n\}$. We use lowercase boldface characters for vectors. We use $\mathbf{x} \cdot \mathbf{y}$ for the inner product of $\mathbf{x}, \mathbf{y} \in \mathbb{R}^d$. For $\mathbf{x} \in \mathbb{R}^d$, $\|\mathbf{x}\|_2$ denotes the $\ell_2$-norm of $\mathbf{x}$. For a set of vectors $k$, $\mathbf{X} = \{\boldsymbol{x}_1, \ldots, \boldsymbol{x}_k\}$, $\mathbf{X}_{i-j}$ denotes the difference between the vectors $\boldsymbol{x}_i, \boldsymbol{x}_j$, i.e., $\mathbf{X}_{i-j} = \boldsymbol{x}_i - \boldsymbol{x}_j$. We use $\mathbb{1}_A = \mathbb{1}\{A\}$ to denote the characteristic function of the set $A$. We use the standard $O(\cdot), \Theta(\cdot), \Omega(\cdot)$ asymptotic notation. We use $\mathbf{E}_{X \sim \mathcal{D}}[X]$ for the expectation of a random variable $X$ according to the distribution $\mathcal{D}$ and $\mathbf{Pr}[\mathcal{E}]$ for the probability of event $\mathcal{E}$. For simplicity of notation, we omit the distribution when it is clear from the context.

## 2. Online Learning with Massart Noise

In this section, we provide an algorithm for online learning halfspaces with Massart noise. The learner iteratively processes a sequence of covariates and associated labels $(\boldsymbol{x}, y)$ provided by the adversary, chooses a label corresponding to the current decision vector $\boldsymbol{w}$, and then updates (even if the label was correct) the decision vector using a carefully chosen convex loss. Our loss is based on a modification of the LeakyReLU loss, i.e., $\text{LeakyReLU}_\lambda(t) \triangleq (1-\lambda)t\mathbb{1}\{t > 0\} + \lambda t\mathbb{1}\{t < 0\}$. First, we show an equivalent expression for the Leaky-ReLU loss.

**Fact 2.1.** *Let* $\text{LeakyReLU}_\lambda(t) \triangleq (1 - \lambda)t\mathbb{1}\{t > 0\} + \lambda t\mathbb{1}\{t < 0\}$. *It can be equivalently expressed as* $\text{LeakyReLU}_\lambda(t) \triangleq 1/2((1 - 2\lambda)|t| + t)$.

*Proof.* Note that it holds $t\mathbb{1}\{t > 0\} = (t + |t|)/2$ and $t\mathbb{1}\{t < 0\} = (t - |t|)/2$. Therefore by plugging these identities in the definition of $\text{LeakyReLU}_\lambda(t)$, we obtain the result. $\square$

In our algorithm, we use as loss the $\text{LeakyReLU}_\lambda(-ty)$ where $t \in \mathbb{R}$ and $y \in \{\pm 1\}$. For the sake of brevity in notation, we define $C_\Delta(t; y) = \text{LeakyReLU}_{(1-\Delta)/2}(-ty)$, i.e.,

$$C_\Delta(t; y) \triangleq (1/2)(\Delta|t| - yt).$$

We provide some intuition behind our choice of the Leaky-ReLU loss. Notice, that in Definition 1.1 we have the label consistency which corresponds to the condition $\mathbf{E}[y]\text{sign}(\boldsymbol{w}^* \cdot \boldsymbol{x}) \geq (1 - 2\eta) := \Delta$. This is clear, from the fact that $y \neq \text{sign}(\boldsymbol{w}^* \cdot \boldsymbol{x})$ with probability at most $\eta$ and that, $\mathbf{E}_y[y] = \mathbf{E}_y[\text{sign}(\boldsymbol{w}^* \cdot \boldsymbol{x})(\mathbb{1}\{y = \text{sign}(\boldsymbol{w}^* \cdot \boldsymbol{x})\} - \mathbb{1}\{y \neq \text{sign}(\boldsymbol{w}^* \cdot \boldsymbol{x})\})] = \text{sign}(\boldsymbol{w}^* \cdot \boldsymbol{x})(1 - 2\mathbf{E}[\mathbb{1}\{y \neq \text{sign}(\boldsymbol{w}^* \cdot \boldsymbol{x})\}])$, therefore $\mathbf{E}[y]\text{sign}(\boldsymbol{w}^* \cdot \boldsymbol{x}) \geq \Delta$. The Leaky-ReLU loss has the property that $\mathbf{E}_y[C_\Delta(-\boldsymbol{w}^* \cdot \boldsymbol{x}; y)] \leq 0$ (see Claim 2.3), which we can later use to design a loss for our problem. However, optimizing this loss function exclusively does not guarantee minimal regret. To this end, we define the loss function $\ell_{\boldsymbol{u}, \tau, \Delta}(\boldsymbol{w}, y, \boldsymbol{x}) := \frac{C_\Delta(\boldsymbol{w} \cdot \boldsymbol{x}; y)}{\max(|\boldsymbol{u} \cdot \boldsymbol{x}|, \tau)}$. To simplify the notation, we define $\ell^{(t)}(\boldsymbol{w}) = \frac{C_{\widetilde{\Delta}}(\boldsymbol{w} \cdot \boldsymbol{x}^{(t)}; y^{(t)})}{\max(|\boldsymbol{w}^{(t)} \cdot \boldsymbol{x}^{(t)}|, \tau)}$, where $\tau, \widetilde{\Delta}$ are fixed and the other parameters are changing through the iterations of our algorithm. Subsequently, we demonstrate that minimizing the regret associated with these reweighted chosen loss functions concurrently yields substantive guarantees for the regret in the context of Definition 1.1.

**Lemma 2.2.** *Assume that a sequence* $\boldsymbol{w}^{(t)}$ *is produced by a (possibly randomized) online algorithm* $\mathcal{A}$ *with the guarantee that* $\mathbf{E}[\sum_{t=1}^T \ell^{(t)}(\boldsymbol{w}^{(t)}) - \sum_{t=1}^T \ell^{(t)}(\boldsymbol{w}^*)] \leq \bar{R}(T)$, *with* $\ell^{(t)}(\boldsymbol{w}) = \frac{C_{\widetilde{\Delta}}(\boldsymbol{w} \cdot \boldsymbol{x}^{(t)}; y^{(t)})}{\max(|\boldsymbol{w}^{(t)} \cdot \boldsymbol{x}^{(t)}|, \tau)}$. *Then, for* $R(T, \epsilon, \gamma, \tau) = T((\epsilon/2) + (8\tau)/(\epsilon\gamma)) + \bar{R}(T)(1 + (8\tau)/(\epsilon\gamma))$, *it holds*

$$\sum_{t=1}^T \mathbf{E}[(\mathbb{1}\{\text{sign}(\boldsymbol{w}^{(t)} \cdot \boldsymbol{x}^{(t)}) \neq y^{(t)}\})] \leq T\eta + R(T, \epsilon, \gamma, \tau).$$

---

**Algorithm 1** Online Learning Massart Halfspaces

1. $\widetilde{\Delta} \leftarrow 1 - 2\eta - \epsilon$, $\tau \leftarrow \epsilon\gamma/4$ and $\boldsymbol{w}^{(0)} = \boldsymbol{e}_1$.
2. For $t = 1, \ldots, T$:
   (a) Adversary selects point $\boldsymbol{x}^{(t)} \in \mathbb{R}^d$ and generates label $y^{(t)}$.
   (b) Learner observes $\boldsymbol{x}^{(t)}$ and chooses label $\hat{y}^{(t)} = \text{sign}(\boldsymbol{w}^{(t)} \cdot \boldsymbol{x}^{(t)})$
   (c) Learner gets label $y^{(t)}$.
   (d) Set
   $$\ell^{(t)}(\boldsymbol{w}) = \frac{C_{\widetilde{\Delta}}(\boldsymbol{w} \cdot \boldsymbol{x}^{(t)}; y^{(t)})}{\max(|\boldsymbol{w}^{(t)} \cdot \boldsymbol{x}^{(t)}|, \tau)}$$
   (e) Run Online Convex Optimization on $\ell^{(t)}(\cdot)$.

---

*Proof.* Denote $\epsilon = \Delta - \widetilde{\Delta}$, with $\epsilon < \Delta/2$ and let $\tau \leq \epsilon\gamma/2$. Let $(\mathcal{F}^{(t)})_{t \in [T]}$ be a filtration adapted to stochastic sequence $\boldsymbol{w}^{(t)}$. We note that the expectations in this proofs are with respect the random variables $y^{(t)}$. First, we show that the optimal decision vector $\boldsymbol{w}^*$ gets a negative loss on expectation.

**Claim 2.3.** *It holds* $\mathbf{E}_{y^{(t)}}[C_{\widetilde{\Delta}}(\boldsymbol{w}^* \cdot \boldsymbol{x}^{(t)}; y^{(t)}) \mid \mathcal{F}^{(t)}] \leq -(\epsilon/2)|\boldsymbol{w}^* \cdot \boldsymbol{x}^{(t)}|$.

*Proof.* Recall that $C_{\widetilde{\Delta}}(t; y) = (1/2)(\widetilde{\Delta}|t| - yt)$, therefore, we have that

$$\mathbf{E}_{y^{(t)}}[C_{\widetilde{\Delta}}(\boldsymbol{w}^* \cdot \boldsymbol{x}^{(t)}; y^{(t)}) \mid \mathcal{F}^{(t)}]$$
$$= \frac{1}{2}(\widetilde{\Delta}|\boldsymbol{w}^* \cdot \boldsymbol{x}^{(t)}| - \mathbf{E}_{y^{(t)}}[y^{(t)} \mid \mathcal{F}^{(t)}]\boldsymbol{w}^* \cdot \boldsymbol{x}^{(t)})$$
$$= \frac{1}{2}(\widetilde{\Delta} - \mathbf{E}_{y^{(t)}}[y^{(t)} \mid \mathcal{F}^{(t)}]\text{sign}(\boldsymbol{w}^* \cdot \boldsymbol{x}^{(t)}))|\boldsymbol{w}^* \cdot \boldsymbol{x}^{(t)}|$$
$$\leq \frac{1}{2}(\widetilde{\Delta} - \Delta)|\boldsymbol{w}^* \cdot \boldsymbol{x}^{(t)}|,$$

where we used that $\mathbf{E}_{y^{(t)}}[y^{(t)} \mid \mathcal{F}^{(t)}]\text{sign}(\boldsymbol{w}^* \cdot \boldsymbol{x}^{(t)}) \geq \Delta$. $\square$

We first show that $\mathbf{E}[\sum_{t=1}^T \ell^{(t)}(\boldsymbol{w}^*)] \leq -T\gamma\epsilon/2$. Using the tower rule, we have that $\mathbf{E}[\sum_{t=1}^T \ell^{(t)}(\boldsymbol{w}^*)] = \sum_{t=1}^T \mathbf{E}[\ell^{(t)}(\boldsymbol{w}^*) \mid \mathcal{F}^{(t)}]$. From Claim 2.3, we have that $\mathbf{E}_{y^{(t)}}[C_{\widetilde{\Delta}}(-\boldsymbol{w}^* \cdot \boldsymbol{x}^{(t)}; y^{(t)})] \leq -(\epsilon/2)|\boldsymbol{w}^* \cdot \boldsymbol{x}^{(t)}|$. Hence, we have that

$$\sum_{t=1}^T \mathbf{E}[\ell^{(t)}(\boldsymbol{w}^*) \mid \mathcal{F}^{(t)}] \tag{1}$$

$$\leq -(\epsilon/2) \sum_{t=1}^T \frac{|\boldsymbol{w}^* \cdot \boldsymbol{x}^{(t)}|}{\max(|\boldsymbol{w}^{(t)} \cdot \boldsymbol{x}^{(t)}|, \tau)}$$

$$\leq -(\epsilon\gamma/2) \sum_{t=1}^T \frac{1}{\max(|\boldsymbol{w}^{(t)} \cdot \boldsymbol{x}^{(t)}|, \tau)}, \tag{2}$$

where we used that $|\boldsymbol{w}^* \cdot \boldsymbol{x}^{(t)}| \geq \gamma$ from the assumptions. Recall that $C_{\widetilde{\Delta}}(\boldsymbol{w} \cdot \boldsymbol{x}; y) = (1/2)(\widetilde{\Delta} - y\mathrm{sign}(\boldsymbol{w} \cdot \boldsymbol{x}))|\boldsymbol{w} \cdot \boldsymbol{x}|$. Let $g^{(t)}(y) = (1/2)(\widetilde{\Delta} - y\mathrm{sign}(\boldsymbol{w}^{(t)} \cdot \boldsymbol{x}^{(t)}))$. Using the tower rule, we get

$$\mathbf{E}\left[\sum_{t=1}^{T} \ell^{(t)}(\boldsymbol{w}^{(t)}) - \sum_{t=1}^{T} \ell^{(t)}(\boldsymbol{w}^*)\right] =$$
$$\sum_{t=1}^{T}\left(g^{(t)}(\mathbf{E}[y^{(t)} \mid \mathcal{F}^{(t)}])\frac{|\boldsymbol{w}^{(t)} \cdot \boldsymbol{x}^{(t)}|}{\max(|\boldsymbol{w}^{(t)} \cdot \boldsymbol{x}^{(t)}|, \tau)}\right.$$
$$\left.- \mathbf{E}[\ell^{(t)}(\boldsymbol{w}^*)]\right) .$$

We define $J$ to be the set of rounds where the $|\boldsymbol{w}^{(t)} \cdot \boldsymbol{x}^{(t)}|$ is smaller than the threshold $\tau$, i.e., $J = \{t : |\boldsymbol{w}^{(t)} \cdot \boldsymbol{x}^{(t)}| \leq \tau\}$. We have that

$$\sum_{t \notin J} g^{(t)}(\mathbf{E}[y^{(t)} \mid \mathcal{F}^{(t)}])\frac{|\boldsymbol{w}^{(t)} \cdot \boldsymbol{x}^{(t)}|}{\max(|\boldsymbol{w}^{(t)} \cdot \boldsymbol{x}^{(t)}|, \tau)}$$
$$= \sum_{t \notin J} g^{(t)}(\mathbf{E}[y^{(t)} \mid \mathcal{F}^{(t)}]) \geq -(T - |J|)/2 , \quad (3)$$

and that $\sum_{t \in J} g^{(t)}(\mathbf{E}[y^{(t)} \mid \mathcal{F}^{(t)}])\frac{|\boldsymbol{w}^{(t)} \cdot \boldsymbol{x}^{(t)}|}{\max(|\boldsymbol{w}^{(t)} \cdot \boldsymbol{x}^{(t)}|, \tau)} \geq -|J|/2$. Moreover note that from Inequality 2, it holds that $\sum_{t \in J} -\mathbf{E}[\ell^{(t)}(\boldsymbol{w}^*)] \geq (\epsilon\gamma)/(2\tau)|J|$ and that $\sum_{t \notin J} -\mathbf{E}[\ell^{(t)}(\boldsymbol{w}^*)] \geq 0$. Therefore, we get that

$$\sum_{t \in J}^{T} \mathbf{E}[\ell^{(t)}(\boldsymbol{w}^{(t)})] - \sum_{t \in J}^{T} \mathbf{E}[\ell^{(t)}(\boldsymbol{w}^*)] \geq |J|\frac{\epsilon\gamma}{2\tau} - |J|\frac{1}{2} ,$$

and by using our assumption that $\tau \leq \epsilon\gamma/2$, we get that $\sum_{t \in J}^{T} \mathbf{E}[\ell^{(t)}(\boldsymbol{w}^{(t)})] - \sum_{t \in J}^{T} \mathbf{E}[\ell^{(t)}(\boldsymbol{w}^*)] \geq |J|(\epsilon\gamma)/(4\tau)$. Using the assumption for the regret guarantee, we have that $\mathbf{E}[\sum_{t=1}^{T} \ell^{(t)}(\boldsymbol{w}^{(t)})] \leq \bar{R}(T) + \mathbf{E}[\sum_{t=1}^{T} \ell^{(t)}(\boldsymbol{w}^*)]$ , which is equivalent to

$$\underbrace{\sum_{i \notin J} \mathbf{E}[\ell^{(t)}(\boldsymbol{w}^{(t)})] \leq \bar{R}(T)}_{I_1}$$
$$-\underbrace{\sum_{i \in J}(\mathbf{E}[\ell^{(t)}(\boldsymbol{w}^{(t)})] - \mathbf{E}[\ell^{(t)}(\boldsymbol{w}^*)])}_{I_2} + \underbrace{\sum_{i \notin J} \mathbf{E}[\ell^{(t)}(\boldsymbol{w}^*)]}_{I_3} .$$
$$(4)$$

Using that $I_1 \geq (|J| - T)/2$ from Inequality 3, that $I_2 \leq -|J|(\epsilon\gamma)/(4\tau)$ from above and that $I_3 \leq 0$ from Claim 2.3, we have that

$$|J| \leq (\bar{R}(T) + T)(4\tau)/(\epsilon\gamma) . \quad (5)$$

Finally, from the regret guarantees, i.e., Inequality 4, we have that $\sum_{t \notin J} \mathbf{E}[\ell^{(t)}(\boldsymbol{w}^{(t)})] \leq \bar{R}(T)$. Recall that that $g^{(t)}(\mathbf{E}[y]) = (1/2)(\widetilde{\Delta} - \mathbf{E}[y]\mathrm{sign}(\boldsymbol{w}^{(t)} \cdot \boldsymbol{x}^{(t)}))$. By adding

$\sum_{t \in J}(\widetilde{\Delta} - \mathrm{sign}(\boldsymbol{w}^{(t)} \cdot \boldsymbol{x}^{(t)}) \mathbf{E}[y^{(t)}])$ on both sides of Inequality 4 and using that $g^{(t)}(\mathbf{E}[y]) \leq 2$ for all $t \in [T]$, we have:

$$\sum_{t \in J}(\widetilde{\Delta} - \mathrm{sign}(\boldsymbol{w}^{(t)} \cdot \boldsymbol{x}^{(t)}) \mathbf{E}[y^{(t)}])$$
$$+ \sum_{t \notin J}(\widetilde{\Delta} - \mathrm{sign}(\boldsymbol{w}^{(t)} \cdot \boldsymbol{x}^{(t)}) \mathbf{E}[y^{(t)}])$$
$$\leq 2\bar{R}(T) + 2|J| , \quad (6)$$

where we have used that $\sum_{t \in J}(\widetilde{\Delta} - \mathrm{sign}(\boldsymbol{w}^{(t)} \cdot \boldsymbol{x}^{(t)}) \mathbf{E}[y^{(t)}]) \leq 2|J|$ and that $I_2, I_3 \leq 0$ as discuseed above. Equivalently, using Inequality 5, we obtain the following bound over all rounds $t = 1, \ldots, T$: $\sum_{t=1}^{T}(\widetilde{\Delta} - \mathrm{sign}(\boldsymbol{w}^{(t)} \cdot \boldsymbol{x}^{(t)}) \mathbf{E}[y^{(t)}]) \leq 2\bar{R}(T) + (\bar{R}(T) + T)4\tau/(\epsilon\gamma)$ . Using that $\mathrm{sign}(\boldsymbol{w}^{(t)} \cdot \boldsymbol{x}^{(t)})y^{(t)} = 1 - 2\mathbb{1}\{\mathrm{sign}(\boldsymbol{w}^{(t)} \cdot \boldsymbol{x}^{(t)}) \neq y^{(t)}\}$ and $\widetilde{\Delta} = 1 - 2\eta - \epsilon$, we get that

$$\sum_{t=1}^{T} \mathbf{E}[(\mathbb{1}\{\mathrm{sign}(\boldsymbol{w}^{(t)} \cdot \boldsymbol{x}^{(t)}) \neq y^{(t)}\})]$$
$$\leq T(\eta + \epsilon) + \bar{R}(T) + 16(\bar{R}(T) + T)\tau/(\epsilon\gamma) . \quad \square$$

Before proceeding with the proof of Theorem 1.9, we make use of the following fact about the online gradient descent on convex functions to bound the $\bar{R}(T)$ of Lemma 2.2.

**Fact 2.4** ((Hazan, 2016))**.** *Let $\ell^{(t)}$ be a convex function and let $D = diam(\mathcal{W})$ and let $G = \max_t \|\nabla\ell^{(t)}(\cdot)\|_2$. Online gradient descent with step size: $\lambda = D/(G\sqrt{T})$ guarantees the following for all $T \geq 1$:*

$$R(T) = \sum_{t=1}^{T} \ell^{(t)}(\boldsymbol{w}^{(t)}) - \min_{\boldsymbol{w}} \sum_{t=1}^{T} \ell^{(t)}(\boldsymbol{w}) \leq GD/3\sqrt{T} .$$

*Proof of Theorem 1.9.* Using Fact 2.4, we get that $\bar{R}(T) = O(\sqrt{T}/\tau)$, therefore from Lemma 2.2, we get that the expected total regret is bound above by $\mathbf{E}[\sum_{t=1}(\mathbb{1}\{\mathrm{sign}(\boldsymbol{w}^{(t)} \cdot \boldsymbol{x}^{(t)}) \neq y^{(t)}\})] \leq \eta T + R(T, \epsilon, \gamma, \tau)$, where $R(T, \epsilon, \gamma, \tau) = (\epsilon/2)T + \bar{R}(T) + (\bar{R}(T) + T)8\tau/(\epsilon\gamma)$ To minimize this quantity, we set $\tau = \Theta(\epsilon^{1+\zeta}\gamma)$ for any $\zeta > 0$ and $\epsilon = \Theta(T^{-1/(4+2\zeta)}/\gamma)$ gives $R(T, \epsilon, \gamma, \tau) \leq T^{3/4 - O(\zeta)}/\gamma$. By taking $\zeta$ close to 0 and get the result. $\square$

## 3. Contextual Bandits with Monotone Rewards

In this section, we describe an algorithm for the setting of Definition 1.5. To this end, we use a generalization of the LeakyLeRU loss we used in Theorem 1.9. This loss is described in Algorithm 2. First we show that our loss satisfies some properties required by our main algorithm to work. The proof can be found at Appendix A.

**Claim 3.1.** *The loss $\ell(\boldsymbol{w})$ computed by Algorithm 2 is convex and $2Mk\max(\Lambda, 1/\rho)$-Lipschitz.*

**Algorithm 2** Compute $G(\boldsymbol{w}; \mathbf{X}, \boldsymbol{v}, \boldsymbol{r}, \alpha)$, where $\boldsymbol{w}$ is the argument and $\mathbf{X}, \boldsymbol{v}, \boldsymbol{r}, \alpha$ are parameters.

1. Generate reward differences and context for every $j \in \{1, \ldots, k\}$: $\boldsymbol{y}_j = \boldsymbol{r}_\alpha - \boldsymbol{r}_j$.

2. If $\boldsymbol{u} = \boldsymbol{0}$: Return the loss

$$\ell(\boldsymbol{w}) = G(\boldsymbol{w}; \mathbf{X}, \boldsymbol{v}, \boldsymbol{r}, \alpha) \triangleq -\Lambda \sum_{j \neq \alpha} \boldsymbol{w} \cdot \mathbf{X}_{\alpha - j} y_j$$

3. Otherwise:

(a) For every $j \in \{1, \ldots, k\}$ set:

$$\boldsymbol{z}_j = \mathbf{X}_{\alpha - j} + \rho \, \mathrm{sign}(\mathbf{X}_{\alpha - j} \cdot \boldsymbol{v})\boldsymbol{v} \|\boldsymbol{v}\|_2^{-1}$$

(b) Return the loss
$$\ell(\boldsymbol{w}) = G(\boldsymbol{w}; \mathbf{X}, \boldsymbol{v}, \boldsymbol{r}, \alpha) \triangleq \sum_{j \neq \alpha} \frac{C_\Delta(\boldsymbol{w} \cdot \boldsymbol{z}_j; y_j)}{|\boldsymbol{v} \cdot \boldsymbol{z}_j|}$$

---

**Algorithm 3** Bandits with Monotone Rewards

1. $\boldsymbol{w}^{(0)} = \boldsymbol{e}_1$.

2. For $t = 1, \ldots, T$:

(a) Adversary picks context $\boldsymbol{X}^{(t)} = (\boldsymbol{x}_1^{(t)}, \ldots, \boldsymbol{x}_k^{(t)})$ and samples reward $\boldsymbol{r}^{(t)} \sim D^{(t)}$.

(b) If $\boldsymbol{w}^{(t)} = \boldsymbol{0}$: Learner picks a uniformly random $\alpha^{(t)}$.

(c) Otherwise: Learner picks $\alpha^{(t)} = \mathrm{argmax}_i \, \boldsymbol{w}^{(t)} \cdot \boldsymbol{x}_i^{(t)}$.

(d) Learner flips a coin $c^{(t)}$ with HEADS probability $q$.

(e) If HEADS:

  i. Learner picks uniformly random action $\beta^{(t)}$.

  ii. Learner gets reward $\boldsymbol{r}^{(t)}(\beta^{(t)})$ and defines the fake reward vector $\widetilde{\boldsymbol{r}}^{(t)}$:

$$\widetilde{\boldsymbol{r}}^{(t)}(i) = \begin{cases} (k-1) \, \boldsymbol{r}^{(t)}(\beta^{(t)}) & \text{if } i = \beta^{(t)} \\ M - \boldsymbol{r}^{(t)}(\beta^{(t)}) & \text{if } i \neq \beta^{(t)}, \end{cases}$$

  iii. Set $\ell^{(t)}(\boldsymbol{w}) = \frac{1}{q} G(\boldsymbol{w}; \mathbf{X}^{(t)}, \boldsymbol{w}^{(t)}, \widetilde{\boldsymbol{r}}^{(t)}, \alpha^{(t)})$.

(f) If TAILS: Learner gets reward $\boldsymbol{r}^{(t)}(\alpha^{(t)})$ and sets $\ell^{(t)}(\boldsymbol{w}) = 0$.

(g) Learner performs Online Convex Optimization with loss $\ell^{(t)}(\cdot)$.

---

**Lemma 3.2.** *Let $\boldsymbol{w}^{(t)}$ be the sequence produced by algorithm Algorithm 3 in the bandit setting of Definition 1.5 with exploration probability q. It holds*

$$\mathbf{E}\left[ \sum_{t=1}^T (G(\boldsymbol{w}^{(t)}; \mathbf{X}^{(t)}, \boldsymbol{r}^{(t)}, \alpha^{(t)}) - G(\boldsymbol{w}^*; \mathbf{X}^{(t)}, \boldsymbol{r}^{(t)}, \alpha^{(t)})) \right]$$
$$\leq O(kM\sqrt{T}\Lambda/q) ,$$

*where the expectation is over the randomness of Algorithm 3 and the randomness of the reward vectors $\boldsymbol{r}^{(t)} \sim D^{(t)}$.*

*Proof.* We first show that for any reward vector $\boldsymbol{r}^{(t)}$ the loss $\ell^{(t)}(\boldsymbol{w})$ that we construct at every round is an unbiased estimate of the corresponding full-information loss function $G(\boldsymbol{w}; \mathbf{X}^{(t)}, \boldsymbol{w}^{(t)}, \boldsymbol{r}^{(t)}, \alpha^{(t)})$. We show the following claim, see Appendix A for the proof.

**Claim 3.3** (Unbiased Loss Estimate). *For any $\boldsymbol{w} \in \mathbb{R}^d$ it holds*

$$\mathbf{E}_{c^{(t)}, \beta^{(t)}}[\ell^{(t)}(\boldsymbol{w}) \mid \mathcal{F}^{(t)}, \boldsymbol{r}^{(t)}] = G(\boldsymbol{w}; \mathbf{X}^{(t)}, \boldsymbol{w}^{(t)}, \boldsymbol{r}^{(t)}, \alpha^{(t)}) .$$

We have that the loss $\ell^{(t)}(\boldsymbol{w}) = G(\boldsymbol{w}; \mathbf{X}^{(t)}, \boldsymbol{w}^{(t)}, \widetilde{\boldsymbol{r}}^{(t)}, \alpha^{(t)})$ constructed at each step $t$ of Algorithm 3 is convex since it is the convex combination of the zero loss and the convex losses $G(\boldsymbol{w}; \mathbf{X}^{(t)}, \boldsymbol{w}^{(t)}, \boldsymbol{r}, \alpha^{(t)})$ (see Claim 3.1) for different reward vector vectors $\boldsymbol{r}$ (each realization of the random variable $\beta^{(t)}$ corresponds to a different reward vector $\boldsymbol{r}$). From Fact 2.4 and Claim 3.1 we have that the sequence $\boldsymbol{w}^{(t)}$ produced by Algorithm 3 achieves expected regret

$$\mathbf{E}\left[ \sum_{t=1}^T (G(\boldsymbol{w}^{(t)}; \mathbf{X}^{(t)}, \boldsymbol{r}^{(t)}, \alpha^{(t)}) - G(\boldsymbol{w}^*; \mathbf{X}^{(t)}, \boldsymbol{r}^{(t)}, \alpha^{(t)})) \right]$$
$$\leq O(kM\sqrt{T}\Lambda/q) . \qquad \square$$

Next, we prove a generalization of Lemma 2.2 for the $k$-arm setting. It shows that minimizing the regret over the loss functions, bounds the expected reward of our setting.

**Lemma 3.4.** *Let $\boldsymbol{w}^{(t)}$ be a stochastic process in $\mathbb{R}^d$ adapted to the filtration $(\mathcal{F}^{(t)})_{t \in \mathcal{T}}$ such that*

$$\mathbf{E}\left[ \sum_{t=1}^T G(\boldsymbol{w}^{(t)}; \mathbf{X}^{(t)}, \boldsymbol{w}^{(t)}, \boldsymbol{r}^{(t)}, \alpha^{(t)}) \right.$$
$$\left. - \sum_{t=1}^T G(\boldsymbol{w}^*; \mathbf{X}^{(t)}, \boldsymbol{w}^{(t)}, \boldsymbol{r}^{(t)}, \alpha^{(t)}) \right] \leq \bar{R}(T) .$$

*Then, for $R(T, \gamma, \Delta, \Lambda, k) = (\bar{R}(T)/k)(1 + 1/(\gamma\Lambda)) + TM/\Lambda$, it holds that*

$$\mathbf{E}\left[ \sum_{t=1}^T \boldsymbol{r}_{\alpha^{(t)}}^{(t)} \right] \geq \mathbf{E}\left[ \sum_{t=1}^T \frac{1}{k} \sum_{i=1}^k \boldsymbol{r}_i^{(t)} \right] + (k-1)/k\Delta T$$
$$- R(T, \gamma, \Delta, \Lambda, k) .$$

*Proof.* We denote as $A^{(t)}$ the event that $\boldsymbol{w}^{(t)} \neq \boldsymbol{0}$ and $\rho = \gamma/2$. We first observe that by adding $\rho \text{sign}(\boldsymbol{w} \cdot \mathbf{X}^{(t)}_{\alpha^{(t)}-j}) \boldsymbol{w}$ to the difference $\mathbf{X}^{(t)}_{\alpha^{(t)}-j}$ we do not affect the choice of the the optimal weight vector $\boldsymbol{w}^*$ and our guess $\boldsymbol{w}^{(t)}$. We observe that $\boldsymbol{w}^* \cdot \boldsymbol{z}^{(t)}_j \text{sign}(\boldsymbol{w}^* \cdot \mathbf{X}^{(t)}_{\alpha^{(t)}-j}) \geq (|\boldsymbol{w}^* \cdot \mathbf{X}^{(t)}_{\alpha^{(t)}-j}| - \rho \boldsymbol{w}^* \cdot \boldsymbol{w}^{(t)}/\|\boldsymbol{w}^{(t)}\|_2) \geq \gamma - \rho \geq 0$, therefore $\text{sign}(\boldsymbol{w}^* \cdot \boldsymbol{z}^{(t)}_j) = \text{sign}(\boldsymbol{w}^* \cdot \mathbf{X}^{(t)}_{\alpha^{(t)}-j})$. For $\boldsymbol{w}^{(t)}$ the similarly note that $\boldsymbol{w} \cdot \boldsymbol{z}^{(t)}_j = \text{sign}(\boldsymbol{w}^{(t)} \cdot \mathbf{X}^{(t)}_{\alpha^{(t)}-j})(|\boldsymbol{w}^{(t)} \cdot \mathbf{X}^{(t)}_{\alpha^{(t)}-j}| + \rho\|\boldsymbol{w}^{(t)}\|_2)$.

First, we show that the optimal decision vector $\boldsymbol{w}^*$ gets negative loss on expectation (see Appendix A).

**Claim 3.5.** *It holds that*

$$\mathbf{E}_y\left[\sum_{t=1}^T \ell^{(t)}(\boldsymbol{w}^*))\right] \leq -k\gamma\Delta \sum_{t=1}^T \mathbb{1}\{(A^{(t)})^c\} .$$

Next, we bound the contribution of the loss of the $\boldsymbol{w}^{(t)}$ (See Appendix A for the proof).

**Claim 3.6.** *It holds that*

$$\mathbf{E}[\sum_{t=1}^T \ell^{(t)}(\boldsymbol{w}^{(t)})] = \sum_{t=1}^T \sum_{j\neq\alpha^{(t)}} (1/2)(\Delta$$
$$- \text{sign}(\boldsymbol{w}^{(t)} \cdot \mathbf{X}^{(t)}_{\alpha^{(t)}-j}) \mathbf{E}[(\boldsymbol{r}^{(t)}_{\alpha^{(t)}} - \boldsymbol{r}^{(t)}_j)])\mathbb{1}\{A^{(t)}\} .$$

Let $J = \sum_{t=1}^T \mathbb{1}\{(A^{(t)})^c\}$, we bound from above $J$ (See Appendix A for the proof).

**Claim 3.7.** *It holds that*

$$J \leq (\bar{R}(T) + TM(k-1))/((k-1)\gamma\Delta\Lambda) .$$

By plugging Inequality 8 and using Claim 3.5 in the assumption for the regret guarantee, we get that

$$\sum_{t=1}^T \sum_{j\neq\alpha^{(t)}} \left( \mathbb{1}\{A^{(t)}\}(\Delta - \text{sign}(\boldsymbol{w}^{(t)} \cdot \mathbf{X}^{(t)}_{\alpha^{(t)}-j}) \right.$$
$$\left. \mathbf{E}[\boldsymbol{r}^{(t)}_{\alpha^{(t)}} - \boldsymbol{r}^{(t)}_j]) \right) \leq 2\bar{R}(T) . \tag{7}$$

Finally, we need to bound from below the term $\mathbf{E}\left[\sum_{t=1}^T \boldsymbol{r}^{(t)}_{\alpha^{(t)}}\right]$. For this reason, we need to connect the reward of each round $\boldsymbol{r}^{(t)}_{\alpha^{(t)}}$ with the regret of the loss (7). Note that if the learner chose the action $\alpha^{(t)}$ that means that $\boldsymbol{w}^{(t)} \cdot \mathbf{X}^{(t)}_{\alpha^{(t)}-j} \geq 0$ for all $j$ given that we are in the event

$A^{(t)}$. Therefore we can decompose $\boldsymbol{r}^{(t)}_{\alpha^{(t)}}$ as follows

$$\boldsymbol{r}^{(t)}_{\alpha^{(t)}} = \frac{1}{k}\sum_{i=1}^k \boldsymbol{r}^{(t)}_i$$
$$+ \frac{1}{k}\sum_{j\neq\alpha^{(t)}}^k \text{sign}(\boldsymbol{w}^{(t)} \cdot \mathbf{X}^{(t)}_{\alpha^{(t)}-j})(\boldsymbol{r}^{(t)}_{\alpha^{(t)}} - \boldsymbol{r}^{(t)}_j) .$$

Furthermore, note that when we are in the event $(A^c)^{(t)}$, the learner chooses a random action, therefore, using Inequality 7 and the equality above, we have that the expected reward is

$$\mathbf{E}\left[\sum_{t=1}^T \boldsymbol{r}^{(t)}_{\alpha^{(t)}}\right] \geq \mathbf{E}\left[\sum_{t=1}^T \frac{1}{k}\sum_{i=1}^k \boldsymbol{r}^{(t)}_i\right]$$
$$+ \frac{k-1}{k}\Delta(T-J) - (2\bar{R}(T)/k) .$$

Using Claim 3.7 and setting $R(T, \gamma, \Delta, \Lambda, k) = (\bar{R}(T)/k)(1 + 1/(\gamma\Lambda)) + TM/\Lambda$, we complete the proof of Lemma 3.4. $\square$

The proof of Theorem 1.11 follows from Lemma 3.4 and Lemma 3.2, see Appendix A for details.

## 4. Conclusions and Open Problems

In this work we considered online linear classification in the Massart or Bounded online classification model of (Ben-David et al., 2009). Under a standard (and necessary) margin assumption, we gave the first efficient algorithm for this problem achieving a mistake bound of $\eta T + o(T)$. This bound is essentially optimal due to known hardness results in the Statistical Query model (Diakonikolas et al., 2022). We extended our online learning setting to a $k$-arm Bandit model that lies between the commonly used regression-based, realizable and the pessimistic agnostic classification contextual bandit models. In this model, we utilized our online Massart learner to obtain an efficient bandit algorithm that obtains roughly $(1-1/k)\Delta T$ more reward than playing at random at every round. We observed that our reduction is tight for the case of 2 arms (given the aforementioned SQ hardness results for learning with Massart noise). However, for $k > 2$ arms it is unclear whether this reward bound is best possible, as the gap between playing at random and playing the best arm at every round may be much larger than $\Delta$. We leave this as an interesting open problem for future work.

## Impact Statement

This paper presents work whose goal is to advance the field of Machine Learning. This work is theoretical in nature and focuses on advancing fundamental knowledge. As such, it

does not directly raise any societal or ethical concerns that warrant special consideration.

## Acknowledgements

ID was supported NSF Medium Award CCF-2107079, ONR award number N00014-25-1-2268, and an H.I. Romnes Faculty Fellowship. VK was supported in part by by NSF Award CCF-2144298. CT was supported by NSF Award CCF-2144298. NZ was supported in part by NSF Medium Award CCF-2107079.

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

# Supplementary Material

# A. Omitted Proofs from Section 3

### A.1. Proof of Claim 3.1

We prove the following:

**Claim A.1.** *The loss $\ell(\boldsymbol{w}) = G(\boldsymbol{w}; \mathbf{X}, \boldsymbol{v}, \boldsymbol{r}, \alpha)$ generated by Algorithm 2 is convex and $2Mk\max(\Lambda, 1/\rho)$-Lipschitz.*

*Proof.* First, $\ell(\cdot)$ is convex because is a sum of convex functions. Moreover, note that the derivative is

$$\nabla \ell(\boldsymbol{w}) = \sum_{j \neq \alpha} \frac{1}{2} \frac{\Delta \text{sign}(\boldsymbol{w} \cdot \boldsymbol{z}_j) - y}{|\boldsymbol{v} \cdot \boldsymbol{z}_j|} \boldsymbol{z}_j \ ,$$

and note that $|y| \leq \max_i |\boldsymbol{r}_i|$ and $\|\boldsymbol{z}_j\|_2 \leq 2 + \rho$ since all the $\boldsymbol{x}_i$ have norm at most 1. Hence, $\|\nabla \ell(\boldsymbol{w})\|_2$ is upper bounded by $k \max_i |\boldsymbol{r}_i| \max_i (1/|\boldsymbol{u} \cdot \boldsymbol{z}_i|) \leq 2k \max_i |\boldsymbol{r}_i|/\rho$. $\qquad\square$

### A.2. Proof of Claim 3.5

We provide proof for the following claim:

**Claim A.2.** *It holds that $\mathbf{E}_y[\sum_{t=1}^{T} \ell^{(t)}(\boldsymbol{w}^*))] \leq -k\gamma\Delta \sum_{t=1}^{T} \mathbb{1}\{(A^{(t)})^c\}$.*

*Proof.* Using the tower rule, we have that

$$\mathbf{E}_y[\sum_{t=1}^{T} \ell^{(t)}(\boldsymbol{w}^*))] = \sum_{t=1}^{T} \mathbf{E}_y[\ell^{(t)}(\boldsymbol{w}^*)) \mathbb{1}\{A^{(t)}\} + \mathbb{1}\{(A^c)^{(t)}\}] \ .$$

We first show that $\mathbf{E}_y[\ell^{(t)}(\boldsymbol{w}^*)) \mathbb{1}\{A^{(t)}\}] \leq 0$. Recall that $C_\Delta(t; y) = \frac{1}{2}(\Delta|t| - yt)$. By taking the expectation over $y$ in the $\mathbf{E}_y[C_\Delta(-\boldsymbol{w}^* \cdot \boldsymbol{z}_j^{(t)}; y_j^{(t)})]$, we get that

$$\mathbf{E}_y[C_\Delta(-\boldsymbol{w}^* \cdot \boldsymbol{z}_j^{(t)}; y_j^{(t)})]$$
$$= \frac{1}{2}(\Delta|\boldsymbol{w}^* \cdot \boldsymbol{z}_j^{(t)}| - \mathbf{E}\left[\left(\boldsymbol{r}_{\alpha^{(t)}}^{(t)} - \boldsymbol{r}_j^{(t)}\right)\right] \boldsymbol{w}^* \cdot \boldsymbol{z}_j^{(t)}) \ .$$

Recall that by the Definition 1.5, we have that $\mathbf{E}[(\boldsymbol{r}_{\alpha^{(t)}}^{(t)} - \boldsymbol{r}_j^{(t)})]\text{sign}(\boldsymbol{w}^* \cdot \mathbf{X}_{\alpha^{(t)}-j}^{(t)}) \geq \Delta$ and as we discussed above $\text{sign}(\boldsymbol{w}^* \cdot \boldsymbol{z}_j^{(t)}) = \text{sign}(\boldsymbol{w}^* \cdot \mathbf{X}_{\alpha^{(t)}-j}^{(t)})$. Therefore we have that

$$\mathbf{E}\left[\left(\boldsymbol{r}_{\alpha^{(t)}}^{(t)} - \boldsymbol{r}_j^{(t)}\right)\right] \boldsymbol{w}^* \cdot \boldsymbol{z}_j^{(t)} \geq \Delta|\boldsymbol{w}^* \cdot \boldsymbol{z}_j^{(t)}| \ ,$$

which gives that $\mathbf{E}_y[C_\Delta(-\boldsymbol{w}^* \cdot \boldsymbol{z}_j^{(t)}; y_j^{(t)})] \leq 0$ and hence $\mathbf{E}_y[\ell^{(t)}(\boldsymbol{w}^*)) \mathbb{1}\{A^{(t)}\}] \leq 0$. Next, with similar arguments as before we have that $\mathbf{E}_y[\ell^{(t)}(\boldsymbol{w}^*)) \mathbb{1}\{(A^c)^{(t)}\}] \leq -k\Lambda\Delta\mathbb{1}\{(A^c)^{(t)}\}$ which completes the proof. $\qquad\square$

### A.3. Proof of Claim 3.6

We prove the following:

**Claim A.3.** *It holds that*

$$\mathbf{E}[\sum_{t=1}^{T} \ell^{(t)}(\boldsymbol{w}^{(t)})] = \sum_{t=1}^{T} \sum_{j \neq \alpha^{(t)}} (1/2)\big(\Delta$$
$$- \text{sign}(\boldsymbol{w}^{(t)} \cdot \mathbf{X}_{\alpha^{(t)}-j}^{(t)}) \mathbf{E}[(\boldsymbol{r}_{\alpha^{(t)}}^{(t)} - \boldsymbol{r}_j^{(t)})]\big)\mathbb{1}\{A^{(t)}\} \ .$$

*Proof.* Note that in the case that the event $(A^c)^{(t)}$ the loss of $\boldsymbol{w}^{(t)}$ is zero. Recall that it holds that $C_\Delta(\boldsymbol{w}^{(t)} \cdot \boldsymbol{z}_j^{(t)}; y_j^{(t)}) = (1/2)(\Delta - y_j^{(t)}\mathrm{sign}(\boldsymbol{w}^{(t)} \cdot \boldsymbol{z}_j^{(t)}))|\boldsymbol{w}^{(t)} \cdot \boldsymbol{z}_j^{(t)}|$. To simplify the notation, let $g_j^{(t)}(y_j^{(t)}) = (1/2)(\Delta - y_j^{(t)}\mathrm{sign}(\boldsymbol{w}^{(t)} \cdot \boldsymbol{z}_j^{(t)}))$. Using the tower rule, we get

$$\mathbf{E}[\sum_{t=1}^T \ell^{(t)}(\boldsymbol{w}^{(t)})]$$
$$= \sum_{t=1}^T \sum_{j \neq \alpha^{(t)}} \frac{g_j^{(t)}(\mathbf{E}[y_j^{(t)}])|\boldsymbol{w}^{(t)} \cdot \boldsymbol{z}_j^{(t)}|}{|\boldsymbol{w}^{(t)} \cdot \boldsymbol{z}_j^{(t)}|} \mathbb{1}\{A^{(t)}\}$$
$$= \sum_{t=1}^T \sum_{j \neq \alpha^{(t)}} g_j^{(t)}(\mathbf{E}[y_j^{(t)}])\mathbb{1}\{A^{(t)}\} \ . \tag{8}$$

Moreover, since it holds that $\mathrm{sign}(\boldsymbol{w}^{(t)} \cdot \boldsymbol{z}_j^{(t)}) = \mathrm{sign}(\boldsymbol{w}^{(t)} \cdot \mathbf{X}_{\alpha^{(t)}-j}^{(t)})$, we have $g_j^{(t)}(\mathbf{E}[y_j^{(t)}]) = (1/2)(\Delta - \mathrm{sign}(\boldsymbol{w}^{(t)} \cdot \mathbf{X}_{\alpha^{(t)}-j}^{(t)}) \mathbf{E}[y_j^{(t)}])$. Hence, we have that

$$g_j^{(t)}(\mathbf{E}[y_j^{(t)}])$$
$$= (1/2)\big(\Delta - \mathrm{sign}(\boldsymbol{w}^{(t)} \cdot \mathbf{X}_{\alpha^{(t)}-j}^{(t)}) \mathbf{E}[(\boldsymbol{r}_{\alpha^{(t)}}^{(t)} - \boldsymbol{r}_j^{(t)})]\big) \ .$$

$\square$

## A.4. Proof of Claim 3.3

We restate and prove the following:

**Claim A.4** (Unbiased Loss Estimate). *For any $\boldsymbol{w} \in \mathbb{R}^d$ it holds*

$$\mathop{\mathbf{E}}_{c^{(t)}, \beta^{(t)}} \left[ \ell^{(t)}(\boldsymbol{w}) \mid \mathcal{F}^{(t)}, \boldsymbol{r}^{(t)} \right] = G(\boldsymbol{w}; \mathbf{X}^{(t)}, \boldsymbol{w}^{(t)}, \boldsymbol{r}^{(t)}, \alpha^{(t)}) \ .$$

*Proof.* Since at every step of Algorithm 3 we construct the loss $G(\boldsymbol{w}; \mathbf{X}^{(t)}, \boldsymbol{w}^{(t)}, \widetilde{\boldsymbol{r}}^{(t)}, \alpha^{(t)})$, using Algorithm 2, we denote by $\widetilde{\boldsymbol{y}}^{(t)} = \widetilde{\boldsymbol{r}}_{\alpha^{(t)}}^{(t)} - \widetilde{\boldsymbol{r}}_j^{(t)}$ the reward difference vector of the adapted reward vector $\widetilde{\boldsymbol{r}}$. Moreover, recall that we denote $y_j^{(t)} = \boldsymbol{r}_{\alpha^{(t)}}^{(t)} - \boldsymbol{r}_j^{(t)}$. Notice that $\widetilde{\boldsymbol{y}}^{(t)}$ is a random variable that depends on the uniformly random action $\beta^{(t)}$. The loss depends on whether the $\boldsymbol{w}^{(t)} = \boldsymbol{0}$ or not so, we consider each case separately. We start with the case where $\boldsymbol{w}^{(t)} \neq \boldsymbol{0}$ and we denote this event as $A^{(t)}$. Recall that $\boldsymbol{z}_j^{(t)} = \mathbf{X}_{\alpha^{(t)}-j}^{(t)} + \rho\,\mathrm{sign}(\mathbf{X}_{\alpha^{(t)}-j}^{(t)} \cdot \boldsymbol{w}^{(t)})\boldsymbol{w}^{(t)}/\|\boldsymbol{w}^{(t)}\|_2$ is the vector containing the context differences as computed in Algorithm 2. Taking the expectation with respect to the random coin flip $c^{(t)}$ we obtain:

$$\mathop{\mathbf{E}}_{c^{(t)}, \beta^{(t)}} \left[ \ell^{(t)}(\boldsymbol{w})\mathbb{1}\{A^{(t)}\} \mid \mathcal{F}^{(t)}, \boldsymbol{r}^{(t)} \right] = q\frac{1}{q} \mathop{\mathbf{E}}_{\beta^{(t)}} [G(\boldsymbol{w}; \mathbf{X}^{(t)}, \boldsymbol{w}^{(t)}, \widetilde{\boldsymbol{r}}^{(t)}, \alpha^{(t)})\mathbb{1}\{A^{(t)}\} \mid \mathcal{F}^{(t)}, \boldsymbol{r}^{(t)}] =$$

$$\mathop{\mathbf{E}}_{\beta^{(t)}} \left[ \sum_{j \neq \alpha^{(t)}} \frac{C_\Delta(\boldsymbol{w} \cdot \boldsymbol{z}_j^{(t)}; \widetilde{\boldsymbol{y}}_j^{(t)})}{|\boldsymbol{w}^{(t)} \cdot \boldsymbol{z}_j^{(t)}|} \mathbb{1}\{A^{(t)}\} \,\middle|\, \mathcal{F}^{(t)}, \boldsymbol{r}^{(t)} \right] = \sum_{j \neq \alpha^{(t)}} \frac{\mathbf{E}_{\beta^{(t)}}[C_\Delta(\boldsymbol{w} \cdot \boldsymbol{z}_j^{(t)}; \widetilde{\boldsymbol{y}}_j^{(t)}) \mid \mathcal{F}^{(t)}, \boldsymbol{r}^{(t)}]}{|\boldsymbol{w}^{(t)} \cdot \boldsymbol{z}_j^{(t)}|} \mathbb{1}\{A^{(t)}\} \ .$$

where for the last equation we used the linearity of expectation and the fact that the action $\alpha^{(t)}$, the event $A^{(t)}$ and the weight vector at $t$-th iteration $\boldsymbol{w}^{(t)}$ do not depend on $\beta^{(t)}$ conditional on $\mathcal{F}^{(t)}$. We now observe that the loss $C_\Delta(t; y) = (1/2)(\Delta|t| - yt)$ is linear in $y$. Therefore, using again the linearity of expectation, we have that

$$\mathop{\mathbf{E}}_{\beta^{(t)}} [C_\Delta(\boldsymbol{w} \cdot \boldsymbol{z}_j^{(t)}; \widetilde{\boldsymbol{y}}_j^{(t)}) \mid \mathcal{F}^{(t)}, \boldsymbol{r}^{(t)}] = C_\Delta(\boldsymbol{w} \cdot \boldsymbol{z}_j^{(t)}; \mathop{\mathbf{E}}_{\beta^{(t)}} [\widetilde{\boldsymbol{y}}_j^{(t)} \mid \mathcal{F}^{(t)}, \boldsymbol{r}^{(t)}]) \ .$$

Next, we consider the case where $\boldsymbol{w}^{(t)} = \boldsymbol{0}$, and we call this event $(A^c)^{(t)}$. We have that by taking the expectation with

respect to the random coin flip $c^{(t)}$ we obtain:

$$\underset{c^{(t)},\beta^{(t)}}{\mathbf{E}}\left[\ell^{(t)}(\boldsymbol{w})\mathbb{1}\{(A^c)^{(t)}\} \mid \mathcal{F}^{(t)},\boldsymbol{r}^{(t)}\right] = \underset{\beta^{(t)}}{\mathbf{E}}[G(\boldsymbol{w};\boldsymbol{X}^{(t)},\boldsymbol{w}^{(t)},\widetilde{\boldsymbol{r}}^{(t)},\alpha^{(t)})\mathbb{1}\{(A^c)^{(t)}\} \mid \mathcal{F}^{(t)},\boldsymbol{r}^{(t)}]$$

$$= \underset{\beta^{(t)}}{\mathbf{E}}[-\sum_{j\neq\alpha^{(t)}}\boldsymbol{w}\cdot\mathbf{X}_{\alpha-j}\widetilde{\boldsymbol{y}}_j^{(t)}\mathbb{1}\{(A^c)^{(t)}\} \mid \mathcal{F}^{(t)},\boldsymbol{r}^{(t)}] = -\sum_{j\neq\alpha^{(t)}}\boldsymbol{w}\cdot\mathbf{X}_{\alpha-j}\underset{\beta^{(t)}}{\mathbf{E}}[\widetilde{\boldsymbol{y}}_j^{(t)} \mid \mathcal{F}^{(t)},\boldsymbol{r}^{(t)}]\mathbb{1}\{(A^c)^{(t)}\},$$

where for the last equation we used the linearity of expectation and the fact that the action $\alpha^{(t)}$ and the event $A^{(t)}$ do not depend on $\beta^{(t)}$ conditional on $\mathcal{F}^{(t)}$.

To finish the proof we have to show that the adapted reward difference vector $\widetilde{\boldsymbol{y}}^{(t)}$ is an unbiased estimate of the true reward difference vector $\boldsymbol{y}^{(t)}$. Since we pick the action $\beta^{(t)}$ uniformly at random from $\{1,\ldots,k\}$, we have that the $i$-th coordinate of the adapted reward vector $\widetilde{\boldsymbol{r}}$ is equal to

$$\underset{\beta^{(t)}}{\mathbf{E}}[\widetilde{\boldsymbol{r}}_i^{(t)} \mid \mathcal{F}^{(t)},\boldsymbol{r}^{(t)}] = \frac{1}{k}(k-1)\boldsymbol{r}_i^{(t)} + \frac{1}{k}\sum_{s\neq i}(M-\boldsymbol{r}_s^{(t)}).$$

Therefore, we have that the expected difference $\widetilde{\boldsymbol{y}}_j^{(t)}$ is equal to

$$\underset{\beta^{(t)}}{\mathbf{E}}[\widetilde{\boldsymbol{y}}_j^{(t)} \mid \mathcal{F}^{(t)},\boldsymbol{r}^{(t)}] = \frac{1}{k}(k-1)\boldsymbol{r}_{\alpha^{(t)}}^{(t)} + \frac{1}{k}\sum_{s\neq\alpha^{(t)}}(M-\boldsymbol{r}_s^{(t)}) - \frac{1}{k}(k-1)\boldsymbol{r}_j^{(t)} - \frac{1}{k}\sum_{s\neq j}(M-\boldsymbol{r}_s^{(t)})$$

$$= \frac{1}{k}(k-1)\boldsymbol{r}_{\alpha^{(t)}}^{(t)} - \frac{1}{k}\boldsymbol{r}_j^{(t)} - \frac{1}{k}(k-1)\boldsymbol{r}_j^{(t)} + \frac{1}{k}\boldsymbol{r}_{\alpha^{(t)}}^{(t)} = \boldsymbol{r}_{\alpha^{(t)}}^{(t)} - \boldsymbol{r}_j^{(t)} = \boldsymbol{y}_j^{(t)}.$$

Therefore, combining the above equations, we conclude that

$$\underset{c^{(t)},\beta^{(t)}}{\mathbf{E}}[\ell^{(t)}(\boldsymbol{w}) \mid \mathcal{F}^{(t)},\boldsymbol{r}^{(t)}]G(\boldsymbol{w};\mathbf{X}^{(t)},\boldsymbol{w}^{(t)},\boldsymbol{r}^{(t)},\alpha^{(t)}).$$

This completes the proof of Claim 3.3. $\qquad\square$

## A.5. Proof of Claim 3.7

We restate and prove the following claim.

**Claim A.5.** *It holds that $J \leq (\bar{R}(T) + TM(k-1))/((k-1)\gamma\Delta\Lambda)$.*

*Proof.* From Inequality 8, we have that

$$\mathbf{E}\left[\sum_{t=1}^T \ell^{(t)}(\boldsymbol{w}^{(t)})\right] = \sum_{t=1}^T\sum_{j\neq\alpha^{(t)}}g_j^{(t)}(\mathbf{E}[y_j^{(t)}])\mathbb{1}\{A^{(t)}\}$$

$$\geq -M(k-1)(T-J)/2 .$$

Furthermore, sing the assumption for the regret guarantee, we have:

$$\mathbf{E}\left[\sum_{t=1}^T \ell^{(t)}(\boldsymbol{w}^{(t)}) - \sum_{t=1}^T \ell^{(t)}(\boldsymbol{w}^*)\right] \leq \bar{R}(T) .$$

Hence, using Claim 3.5, we get the result. $\qquad\square$

## A.6. Proof of Theorem 1.11

We restate and prove Theorem 1.11.

**Theorem A.6** (Monotone $k$-arm Contextual Bandits)**.** *Consider the monotone reward online setting of Definition 1.5. Moreover, for some unit vector $\boldsymbol{w}^* \in \mathbb{R}^d$, assume that for all $t$, it holds that for all $i$ $\|\boldsymbol{X}_i^{(t)}\|_2 \leq 1$ and for all $i \neq j$,*

$|\boldsymbol{w}^* \cdot \mathbf{X}_i^{(t)} - \boldsymbol{w}^* \cdot \mathbf{X}_j^{(t)}|) \geq \gamma$. *There exists a bandit algorithm that runs in* $\mathrm{poly}(d)$ *time per round and selects a sequence of arms* $\alpha^{(1)}, \ldots, \alpha^{(T)} \in [k]$ *that obtain expected reward*

$$\mathbf{E}\left[\sum_{t=1}^{T} \boldsymbol{r}^{(t)}(\alpha^{(t)})\right] \geq \mathbf{E}\left[\sum_{t=1}^{T} \frac{1}{k} \sum_{i=1}^{k} \boldsymbol{r}_i^{(t)}\right]$$
$$+ \frac{k-1}{k}\Delta T - O(T^{5/6}(k\Delta M^2)^{1/3}/\gamma).$$

*Proof of Theorem 1.11.* Algorithm 3 in each iteration, either with probability $q$ makes a random choice or with probability $1 - q$ chooses the best action according to the current decision weight vector (or a random action if $\boldsymbol{w}^{(t)} = \boldsymbol{0}$). Therefore we have that

$$\sum_{t=1}^{T} \mathbf{E}_{c^{(t)}}\left[\boldsymbol{r}^{(t)}\right] = (1-q)\sum_{t=1}^{T} \mathbf{E}\left[\boldsymbol{r}_{\alpha^{(t)}}^{(t)}\right] + \frac{q}{k}\mathbf{E}\left[\sum_{t=1}^{T}\sum_{i=1}^{k} \boldsymbol{r}_i^{(t)}\right].$$

Using Lemma 3.4, we get that

$$\sum_{t=1}^{T} \mathbf{E}\left[\boldsymbol{r}^{(t)}\right] \geq \frac{1}{k}\mathbf{E}\left[\sum_{t=1}^{T}\sum_{i=1}^{k} \boldsymbol{r}_i^{(t)}\right]$$
$$+ (1-q)\frac{k-1}{k}\Delta T - (1-q)R(T, \gamma, \Delta, \Lambda, k).$$

where $R(T, \gamma, \Delta, \Lambda, k) = (\bar{R}(T)/k)(1 + 1/(\gamma\Lambda)) + TM/\Lambda$. From Lemma 3.2 we have that $\bar{R}(T) = (k - 1)M\sqrt{T}/q \max(1/\gamma, \Lambda)$. By maximizing it, we get $\Lambda = 1/\gamma T^{1/6}(M/(k\Delta))^{1/3}$ and $q = M/(\gamma\Lambda\Delta)$. Therefore we get that the expected reward is

$$\sum_{t=1}^{T} \mathbf{E}\left[\boldsymbol{r}^{(t)}\right] \geq \frac{1}{k}\mathbf{E}\left[\sum_{t=1}^{T}\sum_{i=1}^{k} \boldsymbol{r}_i^{(t)}\right]$$
$$+ \frac{k-1}{k}\Delta T - T^{5/6}(k\Delta)^{1/3}M^{2/3}/\gamma.$$

$\square$

