# OpenReview forum: "Online Linear Classification with Massart Noise"
_ICML.cc/2025/Conference — ICML 2025 poster_

### Official Review · Reviewer_qMz2 · 2025-03-11

**Overall Recommendation:** 3

**Summary:**

The paper studies online linear classification under massart noise where noisy label might be flipped with a rate of $\eta$. In the case where the dataset is separable with a margin of $\gamma$. The paper used a scaled leakyRelu function as the surrogate loss and guarantees a mistake bound of $\eta T + O(T^{3/4} / \gamma)$. The binary classification tasks can be extended to $k$-armed bandit setting under the assumption of rewards being monotonic and separated by $\Delta$, an expected reward $(1 - 1/k) \Delta T - o(T)$ is recovered. The later result is a generalization of the first when $k = 2$, with a slight worse dependence on $o(T)$.

**Claims And Evidence:**

The paper claims focusing on computational efficient algorithm with provable guarantee rather than previously established algorithm scales with $O(\sqrt{T})$ but incomputable. The claim is supported by main theorems.

**Essential References Not Discussed:**

well referenced.

**Experimental Designs Or Analyses:**

Not applicable

**Methods And Evaluation Criteria:**

The paper used classical and well acknowledged metric in the filed, mistake bound and reward bounds, for evaluation

**Other Comments Or Suggestions:**

- typo line 234 left '+' instead of '-' followed by $|t|$.

- typo, eqn start from line 312 left: indexing with $t$ instead of $i$

- I did not follow at line 310 right panel, proof to Theorem 1.9. In particular, by definition of $R(T)$ in line 308 right and definition of $\bar R(T)$ in line 268 left. It seems conclude $E[R(T)] \le O(GD / \sqrt{T}) = \bar R(T)$ with appropriate $G, D$ being substituted. What choices allowed the claimed bound $\bar R(T) = O(\sqrt{T} / \tau)$

- Another confusion of above might arise at $\eta$ had clashed interpretation: step size in Fact 2.4 and noise level in Definition 1.1; then $G$ is related to $\eta$ (in the noise sense)...

**Other Strengths And Weaknesses:**

The paper is well written and easy to follow. The intuition of selecting LeakyReul with appropriate parameter is clearly demonstrated.

**Questions For Authors:**

The required assumption in definition 1.5 seems very restrictive if we are considering $k$-armed bandit, as it requires pairwise $\Delta$ separability. $\Delta$ separability between every arm and the best arm is common. I wonder whether definition 1.5 can be lifted to the standard assumption.

**Relation To Broader Scientific Literature:**

the paper contributes to provide new computational efficient online algorithm with theoretical guarantee for linear classification with labels subject to Massart noise. The bound matches with previous offline result.

**Theoretical Claims:**

All checked

---

> ### Author Rebuttal · Authors · 2025-04-01
>
> We thank Reviewer qMz2 for the positive feedback on the paper's writing and clarity of intuition. We will fix the typos pointed out by the reviewer.
>
> **Clarification on $\bar{R}(T)$ Derivation:** We apologize for the confusion regarding the derivation involving $ \bar{R}(T)$ around Line 310. The bound indeed stems from applying Fact 2.4 (Online Gradient Descent regret bound). In our setting, the parameter values are as follows: The domain diameter (D) is equal to 1.
> The gradient norm $G$ is related to the Lipschitz constant of our reweighted loss,
> which is $O(1/\tau)$.
> Substituting these into Fact 2.4 gives the bound $O( \sqrt{T}/\tau)$.
>
> We acknowledge the inconsistent reuse of the symbol $\eta$ for both the Massart noise level and the step size (Fact 2.4). We will change the notation for the step size throughout the paper.
>
> **Restrictiveness of Assumption (Definition 1.5):** The reviewer correctly notes that our assumption of pairwise separability by $\Delta$ for all pairs of arms in the bandit setting is stronger than the more standard assumption requiring only separability between the best arm and all other arms. Relaxing our assumption to the standard "best vs. others" gap introduces significant technical challenges in adapting our Massart noise learner. While this is a very interesting and important direction, we leave it open for future work.

---

### Official Review · Reviewer_WzcY · 2025-03-13

**Overall Recommendation:** 3

**Summary:**

They present a computationally efficient alg that achieves mistake bound \eta T + o(T) where \eta is the probability of flipping the ground truth label in the Massart Noise model. Their algorithm is based on performing online gradient descent on a seq of reweighted Leaky-relU loss functions.

Next, they consider a semi-random k-arm contextual bandit problem, where given a list of contexts, they are consistent with some halfspace $w^*$, in expectation. That is, given a list of contexts $x_1,\cdots, x_k$, given that $w^* x_j > w^* x_i$, the expected reward of action I is larger than action j in expectation by at least \Delta. They used their online Massart learner to obtain an efficient bandit algorithm that obtains roughly (1−1/k)∆T more reward than playing at random at every round. Question: what is a lower bound here, in terms of what is the maximum reward possible?

**Claims And Evidence:**

Their results are definitely novel and interesting, although I did not check all the proofs.

**Essential References Not Discussed:**

I cannot think of anything that is missing.

**Experimental Designs Or Analyses:**

Theoretical paper

**Methods And Evaluation Criteria:**

Theoretical paper, and the claims seem reasonable to me.

**Other Comments Or Suggestions:**

Same as above.

**Other Strengths And Weaknesses:**

Their results are definitely novel and interesting, although I did not check all the proofs. In terms of writing of the paper, at some places, it was very hard to understand what is going on, here are some comments:

Lemma 2.2., you did not define \bar{R}.
Lines 245- 260, what is u? It is not defined.
Alg 2, what is the function G?

Although the results are novel and interesting, I think the write-up can be improved a lot.

**Questions For Authors:**

Same as above.

**Relation To Broader Scientific Literature:**

For the class of \gamma-margin linear classifiers they present the first computationally efficient algorithm that achieves mistake
bound \etaT+o(T).

**Theoretical Claims:**

I did not check all the proofs but the claims seem reasonable to me.

---

> ### Author Rebuttal · Authors · 2025-04-01
>
> We thank Reviewer WzcY for the positive feedback on the paper's novelty. We respond to the reviewer’s questions below.
>
> **$\bar{R}$ in Lemma 2.2**: Thank you for pointing this out. $\bar{R}$ refers to the regret bound used in the online gradient descent analysis.
> Vector $u$ in Lines 245-260: The vector $u$ in Lines 245-260 was meant to represent an auxiliary vector, which is independent of the current vector $w$ in computing the gradients. The value of the vector $u$ is chosen to be the vector $w$ but we define it in this way so that the reweighting does not change the gradient wrt $w$.
>
> **Function $G$ in Algorithm 2**: In Algorithm 2,  $G$ represents the loss function being minimized at each step of Algorithm 2. Its specific form, incorporating the reweighted LeakyReLU loss and bandit feedback, is constructed in steps 2 and 3b of Algorithm 2. We use \ell(w) as a shorthand for the instance of this loss function $G$ at a particular step, simplifying the presentation of the regret analysis.
>
> **Lower Bound Question:**
> The theoretical maximum expected reward achievable by any algorithm (even computationally inefficient ones or those with oracle access to the true separator $w^*$) corresponds to always selecting the optimal arm based on the current context $x^t$. If we assume, without loss of generality, that arm 1 is the optimal arm in expectation for every round $t$ the maximum possible expected reward is $E[\sum_{t=1}^T r_1^t]$.
> Establishing tight bounds on the maximum achievable reward specifically for computationally efficient algorithms in this precise semi-random setting is challenging, and related questions remain open even in offline contexts. For example, in the special case where $k=2$ and $r_i^t\in\{\pm 1\}$, the expected reward gain of $(1/2)\Delta T$ achieved by our algorithm is indeed a meaningful quantity. As discussed in Remarks 1.6 and 1.10, this relates directly to the mistake bound achievable in the underlying binary classification problem. Given the known computational hardness results (as stated in the aforementioned remarks in the submission), the maximum expected reward that can be achieved by computational efficient algorithms is at least $\Delta T$.

---

### Official Review · Reviewer_UTxa · 2025-03-14

**Overall Recommendation:** 3

**Summary:**

This paper considers an online learning setting where context-label pairs are generated with Massart noise.
More specifically, while in standard online classification, both the context and label are generated adversarially, the authors consider a setting where the context is generated adversarially, but the label is determined based on the context at that time and is subject to Massart noise.

They first derive a mistake bound for this online classification setting that matches a computational lower bound in the offline setting.
Technically, they achieve this by considering a new loss function for the online learning algorithm, in which a LeakyReLU function is weighted by the margin of the sample at each time step, which allows them to establish the above mistake bound.

By leveraging this technique, they further derive a desirable mistake bound for the $k$-armed contextual bandit under assumptions weaker than realizability for the reward function.
When the number of arms is 2, this mistake bound matches existing results.

**Claims And Evidence:**

Yes, all propositions and claims in the paper are accompanied by proofs or corresponding references.

**Essential References Not Discussed:**

no

**Ethical Review Concerns:**

a

**Ethics Expertise Needed:**

["Other expertise"]

**Experimental Designs Or Analyses:**

NA

**Methods And Evaluation Criteria:**

Yes, the proposed algorithms are variants of existing algorithms in online classification and are thus valid.
Moreover, the evaluation criteria (mistake bound and expected reward) are standard in the literature.

**Other Comments Or Suggestions:**

no

**Other Strengths And Weaknesses:**

This paper is overall well written.
In particular, the problem setting, algorithms, definitions, and main results are presented in a highly clear manner.
Other strengths are discussed in Relation To Broader Scientific Literature.

A minor weakness is the presence of numerous apparent typos.
For example,
- In Line 72, the phrase "chooses an action $\alpha = 1, \dots, k$" is unclear.
- In Line 80, $\mathbb{E}[r_i | x^{(t)}]$ should be $\mathbb{E}[r_i | x_i^{(t)}]$.

Additionally, some notations are quite confusing.
For example,
- In Fact 2.1, the expression $1/2 ((1-2\lambda)|t| - t)$ is difficult to interpret.
- In Fact 2.4, the regret upper bound $GD/3 \sqrt{T}$ is presented in a fraction format that makes it unclear.

It would be desirable to revise these points for clarity.

**Questions For Authors:**

no

**Relation To Broader Scientific Literature:**

In the traditional online (linear) classification setup, the setting where both the context and label are generated adversarially has been extensively studied.
However, the traditional assumption that the label is generated completely independently of the context is overly pessimistic.
To address this issue, the authors introduce the Massart noise assumption, which has been well-studied in the offline setting, and consider a scenario where there is some dependence between the context and the label, and this is an interesting aspect of this paper.
Furthermore, instead of directly using the LeakyReLU function, which is known to be effective in existing studies, they consider a variant weighted by the margin of the sample at each time step.
By doing so, they establish a mistake upper bound that matches a computational lower bound, and this is of interest to the community.

**Theoretical Claims:**

Yes, I have reviewed the proofs in the main body and confirmed their validity with no issues.

---

> ### Author Rebuttal · Authors · 2025-04-01
>
> We thank the reviewer for the comments. We will fix the typos and address the reviewer’s suggestions to improve clarity.

---

### Official Review · Reviewer_WUUa · 2025-03-14

**Overall Recommendation:** 3

**Summary:**

The paper studies online linear classification with massart noise. The paper designs computationally efficient algorithms for online linear classification with Massart noise. The paper also extends this model to k-arm contextual badntit setting.

## update after rebuttal
After rebuttal, I maintain my score.

**Claims And Evidence:**

In general, the claims are supported by sufficient evidence. The theoretical results seem sound.

**Essential References Not Discussed:**

I don’t know any specific part of the literature that is not addressed.

**Experimental Designs Or Analyses:**

The paper does not include experiments.

**Methods And Evaluation Criteria:**

The performance measure, the mistake bound, makes sense as it is the measure that is studied in the most similar works.

**Other Comments Or Suggestions:**

There is a typo: 234:  |t| - t  ->  |t| + t.

**Other Strengths And Weaknesses:**

1. The paper is well written.
2. This paper gave the first efficient algorithm achieving a mistake bound of $\eta T+o(T)$ in online linear classification with Massart Noise.

**Questions For Authors:**

1. Could you explain why you define a loss function step d in algorithm1, what is the problem of using LeakyReLU directly?

**Relation To Broader Scientific Literature:**

The contributions of this paper fall within the broader field of Online Linear Classification under Massart noise. The most relevant prior work is by Ben-David et al. (2009), who proposed inefficient algorithms. By contrast, this paper presents efficient algorithms.

**Theoretical Claims:**

I checked the general soundness and read all the proofs.

---

> ### Author Rebuttal · Authors · 2025-04-01
>
> We thank Reviewer WUUa for the positive feedback on the clarity and soundness of our theoretical results. We will fix the typos pointed out. In response to the reviewer’s question about the use of the Leaky ReLU:
>
> The standard LeakyReLU loss penalizes points that are further away from the decision boundary more than the points that are close. By reweighting with the margin we equalize the contribution of all points to the objective leading to minimizing misclassification error. Making the reweighting depend on the current vector w would result in a non-convex objective but treating it as a constant everytime results in an online sequence of convex objectives and we prove that our method converges to the desired solution. This is explained in lines 181-191 (left column) of the submission.

---

### Decision · Program_Chairs · 2025-05-01

**Decision:**

Accept (poster)

**Comment:**

The paper studies online linear classification and contextual bandits in the presence of Massart noise. In this case, the paper gives an algorithm that is o(T) away from the best possible offline algorithms. There are a few weaknesses, such as the lack of a discussion on possible lower bounds, and a further comparision with more recent offline works (some of which the authors say are inspired by this paper).

The authors should take into account the feedback from the reviewers and rebuttal in preparing the revision.